# A Unique, Porous C_3_N_4_ Nanotube for Electrochemiluminescence with High Emission Intensity and Long-Term Stability: The Role of Calcination Atmosphere

**DOI:** 10.3390/molecules27206863

**Published:** 2022-10-13

**Authors:** Bolin Zhao, Xingzi Zou, Jiahui Liang, Yelin Luo, Xianxi Liang, Yuwei Zhang, Li Niu

**Affiliations:** School of Civil Engineering c/o Center for Advanced Analytical Science, School of Chemistry and Chemical Engineering, Guangzhou Key Laboratory of Sensing Materials & Devices, Guangzhou University, Guangzhou 510006, China

**Keywords:** electrochemiluminescence, C_3_N_4_ nanotube, Cu^2+^ detection

## Abstract

Developing excellent strategies to optimize the electrochemiluminescence (ECL) performance of C_3_N_4_ materials remains a challenge due to the electrode passivation, causing weak and unstable light emission. A strategy of controlling the calcination atmosphere was proposed to improve the ECL performance of C_3_N_4_ nanotubes. Interestingly, we found that calcination atmosphere played a key role in specific surface area, pore-size and crystallinity of C_3_N_4_ nanotubes. The C_3_N_4_ nanotubes prepared in the Air atmosphere (C_3_N_4_ NT-Air) possess a larger specific surface area, smaller pore-size and better crystallinity, which is crucial to improve ECL properties. Therefore, more C_3_N_4_^•−^ excitons could be produced on C_3_N_4_ NT-Air, reacting with the SO_4_^•−^ during the electrochemical reaction, which can greatly increase the ECL signal. Furthermore, when C_3_N_4_ nanotube/K_2_S_2_O_8_ system is proposed as a sensing platform, it offers a high sensitivity, and good selectivity for the detection of Cu^2+^, with a wide linear range of 0.25 nM~1000 nM and a low detection limit of 0.08 nM.

## 1. Introduction

In recent years, electrochemiluminescence (ECL), a light emission phenomenon controlled by electrochemical oxidation or reduction luminophores, has been widely applied in many fields such as clinical diagnosis, immunoassay, food safety and environmental monitoring, bioanalysis and so on, due to its high sensitivity, wide linear range, low background, and low cost [1,2,3]. In recent years, various nanomaterials have been reported as novel ECL luminophores due to their excellent optical, electrical and chemical properties, including carbon dots, graphene quantum dots, polymer dots, noble metal nanoclusters, carbon nitride (C_3_N_4_), up-conversion materials and so on [4,5,6,7,8,9]. Among them, C_3_N_4_, as a classic ECL emitter materials, has attracted considerable interest due to its high chemical and thermal stability, nontoxicity, low cost and facile synthesis [10,11,12,13]. Nevertheless, the pristine C_3_N_4_ material usually suffers from the problem of weak and unstable ECL signals caused from electrode passivation [14,15]. To solve this problem, Chi’s group loaded the Au nanoparticles on the C_3_N_4_ to trap and store the over-injection electrons in the conduction band of C_3_N_4_, thus preventing the passivation of electrode and obtaining the strong and stable ECL signal [14]. Lu’ group developed a nitrogen vacancy engineering strategy to improve the ECL intensity and stability, and they found that the presence of nitrogen vacancy can facilitate electron transfer and trap the excess of electrons [16]. In a previous study, our group also introduced a one-dimensional C_3_N_4_ nanotube as the ECL luminophore; thanks to the high specific surface area and rapid electron transfer rate of the tubular structure, it displayed a high ECL efficiency and satisfactory stability [17]. Nevertheless, up to now, the strategies for improving the ECL properties of C_3_N_4_ materials are still limited, thus it is urgent to develop new methods to increase the of ECL signal and improve the stability of C_3_N_4_.

Synthesis atmosphere has a great influence on the performance of C_3_N_4_ nanomaterials. For example, Praus’s group prepared the bulk C_3_N_4_ by heating melamine in different atmospheres (Air and N_2_), and found the effect of present oxygen on the photocatalytic activity [18]. Keller’s group also discovered that the NH_3_ played an important role in photocatalytic H_2_ production by varying the g-C_3_N_4_ synthesis atmosphere (air, N_2_, H_2_, Ar and NH_3_) [19]. These reaction conditions usually can change the C/N ratio, crystallinity, band gap, porosity, specific surface area and so on, thus further affect the catalytic performance [20,21,22]. Therefore, the impacts of different calcination atmosphere of C_3_N_4_ on the ECL properties was studied in detail here. 

In this work, we report a method to synthesize the C_3_N_4_ nanotube by a one-step thermal treatment of urea and melamine mixtures in different calcination atmospheres. The derived C_3_N_4_ nanotubes were characterized by multiple techniques, and we found the change of calcination atmosphere will greatly affect the specific surface area, pore-size and the crystallinity of the samples. The C_3_N_4_ nanotube prepared in the air presented the larger specific surface area, smaller pore-size and better crystallinity, which were very helpful for increasing the electrocatalytic active surface area and the number of C_3_N_4_^•−^ excitons, thus improving the ECL performance. Additionally, a highly sensitive and selective ECL method was therefore developed for the sensing of Cu^2+^ based on its interaction with the C_3_N_4_ NT-Air/K_2_S_2_O_8_ system. The obtained results indicated that the proposed ECL sensor showed high sensitivity and good selectivity, which provides a new promising platform for the determination of heavy metal ions. 

## 2. Results and Discussion

### 2.1. Morphology Characterization of C_3_N_4_ Nanotubes

The C_3_N_4_ nanotube were successfully synthesized by simple thermal-polymerization of a mixture of urea and melamine powders at 550 °C in different atmospheres (Air, Ar, N_2_) and the schematic illustration is displayed in Figure 1A. The morphology and microstructure of the three kinds of C_3_N_4_ nanotube samples were first investigated by SEM and TEM images. As shown in Figure 1B, the C_3_N_4_ prepared in the air (C_3_N_4_ NT-Air) possesses a typical tubular structure and the surface of the tube is very rough with a lot of small pores (Appendix A). Overall, these C_3_N_4_ nanotubes were not uniform in length and diameter, having diameters of several hundreds of nanometers and lengths of several micrometers. Meanwhile, these tubes stick to each other to form an interlaced network, which will facilitate the electronic transmission [23,24]. From the TEM images in Figure 1C, the hollow and porous structure of the tube can be seen more clearly and intuitively. The diameter and the thickness of nanotube was about 350 nM and 50 nM, respectively. The hollow tubular structure was maybe caused from the Ostwald Ripening phenomenon during the thermal polymerization process, while the pores on the surface probably attribute to the additional gas release in the heating process [25,26]. These hollow and porous structures generally contribute to expose considerable catalytic active sites and promote the electrolyte diffusion, thus improving the catalytic performance of the material [27,28]. The energy-dispersive X-ray spectroscopy (EDS) elemental mapping of the C_3_N_4_ NT-Air demonstrates that C and N elements are homogeneously distributed over the tubular structure (Figure 1D–F), and the contents of the C and N are 69.49% and 27.97% from the EDS test (Appendix A). Besides, the SEM and TEM images of the C_3_N_4_ calcined under the Ar (C_3_N_4_ NT-Ar) and N_2_ (C_3_N_4_ NT-N_2_) atmospheres are shown in Appendix A. It can be seen that they also keep the tubular structure and are similar to nanotubes prepared in the air, which implies that the change of calcination atmosphere has little effect on the morphology of the samples.

Furthermore, the nitrogen adsorption–desorption test was performed to examine the special surface area and porous structure of the C_3_N_4_ nanotubes. As shown in Figure 2A, the N_2_ adsorption–desorption isotherms of the three samples are similar and they all show typical IV-type isotherms, suggesting the presence of mesoporous in the samples [29]. The specific surface area of the C_3_N_4_ NT-Air, C_3_N_4_ NT-Ar, and C_3_N_4_ NT-N_2_ was calculated to be 58.14, 36.67, and 40.19 m^2^ g^−1^, respectively. The larger surface area for C_3_N_4_ NT-Air sample maybe caused from the more complete aggregation when calcined in the air. According to the pore size distribution in Figure 2B, there are lots of mesopores with a size about 3 nm in the C_3_N_4_ NT-Air sample, while the pore size in C_3_N_4_ NT-Ar and C_3_N_4_ NT-N_2_ are mainly focused on tens of nanometers. According to the previous reports, a smaller pore-size is usually beneficial for improved catalytic performance [30]. So, the increased BET surface area and the decreased pore volume of the C_3_N_4_ NT-Air sample will greatly promote the transmission of reactants and products during the electrochemical reaction process, and improve the performance of adsorption and catalysis [31,32].

### 2.2. Structure and Chemical Compositions of C_3_N_4_ Nanotubes

The crystal structure of as-synthesized samples was studied by XRD analysis. As shown in Figure 3A, the XRD patterns of the three C_3_N_4_ samples all exhibit two characteristic diffraction peaks at 13.1° and 27.4°. The weaker peaks (100) at 13.1° are indexed to the in-plane structural packing motif and the stronger peak (002) at 27.4° corresponds to the interlayer stacking of aromatic systems [33,34]. It should be noted that the diffraction peak intensity of the C_3_N_4_ NT-Air was stronger than others, which implying that the air is conducive to the growth of crystals [35]. The chemical composition of C_3_N_4_ nanotubes was characterized by FT-IR spectroscopy and it can be seen that all the samples show similar absorption bands (Figure 3B). The intense breathing vibration at 810 cm^−1^ is attributed to the tri-s-triazine rings and the dense peaks in the region 1200–1700 cm^−1^ are assigned to the aromatic CN heterocyclic units [36]. The broad absorption peak located at 3000–3600 cm^−1^ is assigned to the N-H stretching vibration of the uncondensed amino groups and O-H of surface-bonded H_2_O molecules [34]. Electron paramagnetic resonance (EPR) tests were also employed to verify the improvement of the crystallinity in the C_3_N_4_ NT-Air sample (Figure 3C). Apparently, C_3_N_4_ NT-Air exhibits much weaker EPR signal, indicating the reduction of defect degree in C_3_N_4_ NT-Air [37,38,39]. XPS survey was conducted to further investigate the elemental composition and surface electronic states of the C_3_N_4_ nanotube materials. From the XPS survey spectra in Figure 3D, it can be seen that all the C_3_N_4_ nanotubes are mainly composed of C and N elements, while the O element is introduced by the physically adsorbed water [24]. The C1s high-resolution spectrum in Figure 3E can be fitted into three peaks, corresponding to the C=C bond (284.8 eV), C-NH_2_ bond (286.5 eV) and N=C-N bond (288.3 eV), respectively [24,28]. Further analysis showed that the proportion of C-NH_2_ in C_3_N_4_ NT-Air was low, which was probably due to the uncondensed -NH groups that easily drop out and turn into NH_3_ when prepared in the air atmosphere. Figure 3F exhibits the high-resolution spectrum of N1s in C_3_N_4_ nanotube, which also can be deconvoluted into three peaks at 398.8 eV, 400.2 eV, 401.4 eV, which can be ascribed to C=N-C, N-C_3_ and C-N-H, respectively [40,41]. The ratio of C-N-H peaks also was also low, similar to the C-NH_2_; these results further indicate the improvement of crystallinity.

### 2.3. ECL Performance of C_3_N_4_ Nanotubes

The ECL and electrochemical behaviors of the C_3_N_4_ nanotubes prepared in different atmospheres were investigated with a cyclic voltammogram (CV) in 0.1 M PBS solution (pH 7.4) under the potential range between 0 and −1.3 V [42]. The C_3_N_4_ nanotube modified GCE electrodes were first tested in the PBS without the K_2_S_2_O_8_ coreactant, and there was only a very weak ECL signal (Appendix A). However, it can be seen from Figure 4A that the ECL emission increased obviously when the 100 mM K_2_S_2_O_8_ coreactant was added into the PBS solution. Moreover, it can be seen that the ECL signal of C_3_N_4_ NT-Air is about 9.1-fold and 5.3-fold higher than the C_3_N_4_ NT-Ar and C_3_N_4_ NT-N_2_, implying that the change of calcination atmosphere has great influence on ECL performance. Meanwhile, the bare GCE was also tested in the PBS with 100 mM K_2_S_2_O_8_, and there was almost no ECL signal that could be found, which confirmed that the luminescence was not caused by the K_2_S_2_O_8_ coreactant (Appendix A). The corresponding CV curves were also studied under the same test condition. Appendix A shows that the bare GCE electrode has an obvious reductive peak at −0.99 V in PBS containing 100 mM K_2_S_2_O_8_, which is perhaps due to the reduction of K_2_S_2_O_8_ [43]. While the three kinds of modified C_3_N_4_ GCE nanotubes in Figure 4B displayed similar curves in the test (whether the potential or the current), this could be ascribed to the electrocatalytic reduction of K_2_S_2_O_8_ reaching its saturation state. 

In order to clarify the probable enhancement mechanism about the ECL performance of the C_3_N_4_ NT-Air sample, the CV curves of C_3_N_4_ NT-Air/GCE, C_3_N_4_ NT-Ar/GCE and C_3_N_4_ NT-N_2_/GCE were further studied in the PBS without K_2_S_2_O_8_. As shown in Figure 4C, C_3_N_4_ NT-Air/GCE shows a much larger reduction current than other two electrodes, which means more electrons will be injected into the conduction band of C_3_N_4_ nanotube and thus more C_3_N_4_^•−^ excitons can be produced during the cathodic reduction process. Then, the C_3_N_4_^•−^ excitons will react with the excess SO_4_^•−^ to produce a stronger light. Therefore, these results mean that the reduction step of the C_3_N_4_ nanotube becomes the key factor affecting the ECL intensity. Besides the excellent ECL emission intensity, long-term stability is also another essential factor to assess the performance of the ECL system in practical applications. Figure 4D shows the outstanding ECL stability of the C_3_N_4_ NT-Air under continuous cyclic scans of 20 cycles, which is crucial for the sensing application. The superior stability of the C_3_N_4_ nanotube can be attributed to the high specific surface area and rapid electron transfer rate of the tubular structure as our previously report [17].

### 2.4. ECL Mechanism of C_3_N_4_ Nanotubes

To further demonstrate the ECL reaction mechanism of the C_3_N_4_ nanotube/K_2_S_2_O_8_ system, the optical properties of C_3_N_4_ nanotubes were characterized by Photoluminescence (PL) and ECL spectra [44]. As shown in Figure 5A, C_3_N_4_ NT-Air displayed a PL emission peak at 444 nm and a shoulder peak at 466 nm, while the C_3_N_4_ NT-Ar and C_3_N_4_ NT-N_2_ showed only one peak at 465 nm and 468 nm, respectively. The red-shift of the PL emission in C_3_N_4_ NT-Ar and C_3_N_4_ NT-N_2_ was maybe due to the existence of defects, which can reduce the band gap in the samples. This also can be verified by the UV-vis DRS spectra and band gap calculation energies in Appendix A. The corresponding ECL spectra of the three kinds samples are shown in Figure 5B. It was observed that the ECL spectra matched well with the PL spectra of C_3_N_4_ nanotube with a slight red-shift of several nanometers. This indicated that the light emission generated from electrochemical reactions was similar to that by photoexcitation, thus the ECL emission of C_3_N_4_ nanotube could be caused by band gap luminescence rather than surface state emission processes [45]. Therefore, the possible reaction mechanism is shown in Figure 5C according to the electrochemistry and optical investigation. During the electrochemical reduction process, the S_2_O_8_^2−^ was firstly reduced to produce strongly oxidizing SO_4_^•−^ (Equation 1) at a low potential, and then the electrons were injected into the conduction band (CB) of C_3_N_4_ nanotube to form the negatively charged C_3_N_4_^•−^ (Equation 2). Subsequently, the SO_4_^•−^ radicals will serve as a hole donor to react with the C_3_N_4_^•−^ and form the excited-state C_3_N_4_^*^ (Equation 3). At last, the excited-state C_3_N_4_^*^ will decay to the ground state, accompanied by the production of blue light. This process is also described as follows:

### 2.5. Detection of Cu^2+^ Ion

In order to evaluate the analytical performance of the ECL system of C_3_N_4_ NT-Air/K_2_S_2_O_8_ couple, the Cu^2+^ heavy metal ion was chosen as an analytical model due to the quenching effect on ECL emission. The ECL signal responses of C_3_N_4_ NT-Air/K_2_S_2_O_8_ with different concentrations of Cu^2+^ are shown in Figure 6A. It can be seen that the ECL intensity gradually decreased with the continuous addition of Cu^2+^ from 0.25 nM to 1000 nM. Figure 6B exhibited the corresponding linear relationship between the change of ECL intensity and the logarithm of Cu^2+^ concentrations. Interestingly, there are two different linear regression equations, Δ*I* = 11,658.1 lg(c/nM) + 10,089.7 (R^2^ = 0.9978) and Δ*I* = 145,314.2 l g(c/nM) − 202,150.7 (R^2^ = 0.9991), which correspond to the concentration range from 0.25~20 nM and 50~1000 nM, respectively, where Δ*I* represents the change of ECL intensity, and c stands for the concentration of Cu^2+^. Moreover, the limit of detection (LOD) was estimated to be 0.08 nM (S/N = 3). All these results demonstrate that this C_3_N_4_ NT-Air/K_2_S_2_O_8_ ECL system has high sensitivity and wide linear range, which is superior to other reported ECL methods for Cu^2+^ detection (Appendix A) [46,47,48,49,50,51,52,53,54].

Furthermore, the stability of the proposed ECL sensor was also investigated under continuous CV scans for 20 cycles with 500 nM Cu^2+^ in PBS (Figure 6C). It was observed that there was no obvious ECL signal fluctuation, which shows a reliable stability of the proposed sensor. On the other hand, the selectivity of the prepared C_3_N_4_ NT-Air ECL sensor was further evaluated using the common metal ions (Co^2+^, Ni^2+^, Mg^2+^, Cr^2+^, Zn^2+^, Mn^2+^, K^+^, Na^+^, Fe^2+^, Pb^2+^, Fe^3+^, Al^3+^) as the interfering agents (Figure 6D). Here, the concentration of the Cu^2+^ was 1.0 μM and the other metal ions’ concentrations were 20 μM; Δ*I* is the change of ECL intensity and *I*_0_ is the initial ECL intensity, respectively. As can be seen from Figure 6D, the interference metal ions show negligible quenching effects on the ECL signals, suggesting the satisfactory selectivity towards Cu^2+^. The reasons for the ECL quenching by Cu^2+^ maybe due to the redox potential of Cu^2+^/Cu^+^ (0.159 V vs. NHE) lying between the conduction and valence bands of C_3_N_4_, which will cause the electron transfer from the C_3_N_4_ nanotube to Cu^2+^ and consume a part of SO_4_^•−^ and then result in ECL quenching [44,55], while the redox potentials of other metal ions are more negative than Cu^2+^, so the electron transfer between them and C_3_N_4_ nanotube is more passive [54]. All these results indicate that the C_3_N_4_ NT-Air/K_2_S_2_O_8_ ECL system has promising sensing applications for the analysis of Cu^2+^.

## 3. Materials and Methods

### 3.1. Materials

Urea (CH_4_N_2_O), melamine (C_3_H_6_N_6_) and potassium peroxodisulfate (K_2_S_2_O_8_) were purchased from Aladdin Co. Ltd (Shanghai, China). Potassium chloride (KCl), disodium hydrogen phosphate (Na_2_HPO_4_), monopotassium phosphate (KH_2_PO_4_) and cupric nitrate (Cu(NO_3_)_2_) were provided by Sinopharm Chemical Reagent Co., Ltd. (Shanghai, China). Nafion solution (5 wt%) was obtained from Sigma-Aldrich Co., Ltd. (St. Louis, MO, USA). All reagents in this work were of analytical grade and used as received without further purification. The ultrapure water (resistivity of 18.25 MΩ cm) used throughout all experiments was purified through a Millipore-Q water purification system.

### 3.2. Preparation of C_3_N_4_ Nanotube

The C_3_N_4_ nanotube was synthesized by the thermal treatment of the urea and melamine mixture according to the previous report with some modifications [56]. First, urea (5 g) and melamine (0.5 g) in a mass ratio pf 10:1 were thoroughly mixed by grinding treatments for 15 min. Then, the mixture was placed into the aluminum crucible and annealed in a tube furnace at 550 °C with a ramp of about 5 °C/min and held for 4 h under different atmosphere (Air, N_2_ and Ar). After the system naturally cooled to room temperature, the resulting yellow products were carefully ground into powders in an agate mortar for further use. 

### 3.3. Preparation of C_3_N_4_ Nanotube Modified Glassy Carbon Electrode

Prior to use, the glassy carbon electrode (GCE, φ = 4mm) was polished carefully with 0.3 and 0.05 μm alumina slurry, respectively, followed by ultrasonic cleaning in water and drying in N_2_ stream. Then, 10 μL of the as-prepared C_3_N_4_ nanotube solution (0.5 mg/mL) was dropped onto the surface of the pretreated GCE and dried at room temperature. Lastly, 5 μL Nafion solution (0.05 wt%) was dropped on the surface of the modified GCE to prevent the C_3_N_4_ nanotube from falling off.

### 3.4. ECL, CV

The ECL measurements were performed on the MPI-EⅡ ECL multifunctional analyzer system (Xi’an Remax Analysis Instrument Co., Ltd., Xi’an, China) in a typical three-electrode setup using the C_3_N_4_ nanotube modified GCE as the working electrode, Pt wire as the counter electrode and Ag/AgCl (with saturated KCl solution) electrode as the reference electrode, respectively. All the ECL signals were recorded in 0.01 M phosphate-buffered solution (PBS, pH 7.4) solution containing 0.1 M K_2_S_2_O_8_ under a continuous potential scan between 0 and −1.3 V with a scanning rate of 0.5 V/s (unless otherwise specified), and the voltage of the photomultiplier tube (PMT) was set at 500 V. 

The cyclic voltammetry (CV) tests were carried out using a CHI 660B electrochemical workstation with the same conditions as the ECL test.

### 3.5. Sample Characterization 

The crystalline structure of the prepared samples was analyzed by an X-ray diffraction (XRD, Bruker D8 ADVANCE) with Cu-Kα radiation (1.54178 Å). The morphologies of the samples were determined by scanning electron microscopy (SEM, JSM-7001F) and transmission electron microscopy (TEM, JEM-2100F). The energy dispersive x-ray spectroscopy (EDX) used for elemental mapping comprised the use of the X-Max 100 microanalysis system from Oxford. Surface chemical compositions and chemical status of the as-prepared materials were analyzed by X-ray photoelectron spectroscopy (XPS, Thermo SCIENTIFIC ESCALAB 250Xi) with a monochromatic Al Kα X-ray source (1487.20 eV). Fourier transform infrared (FT-IR) spectroscopy was recorded on a Bruker Vertex 70 spectrometer. Ultraviolet-visible (UV-vis) absorption spectra were acquired by a UV-1780 spectrophotometer (Shimadzu, Tokyo, Japan). Photoluminescence (PL) spectra were measured at room temperature on an Edinburgh fluorescence spectrometer FLS1000 system. N_2_ adsorption-desorption isotherms, Brunauer-Emmett-Teller (BET) surface area, and pore size distribution were obtained using a MicroActive for ASAP 2460 system at liquid N_2_ temperature.

## 4. Conclusions

In conclusion, taking advantage of different atmospheres, a novel strategy is presented to enhance the ECL intensity and stability of C_3_N_4_ nanotubes. Benefiting from the larger specific surface area, unique porous structure and better crystallinity, the C_3_N_4_ nanotube prepared in air atmosphere shows a stronger ECL signal than the other two samples prepared in Ar and N_2_ (9.1-fold and 5.3-fold). Moreover, the ECL mechanism of C_3_N_4_ NT-Air was proposed so that more C_3_N_4_^•−^ excitons on the surface of C_3_N_4_ NT-Air could be produced to react with the SO_4_^•−^, emitting a stronger light signal. Based on the excellent ECL performance of C_3_N_4_ NT-Air/K_2_S_2_O_8_ system, a high sensitivity and excellent selectivity ECL sensor for Cu^2+^ was successfully developed, which possess a linear response over a range from 0.25~1000 nM and a low detection limit of 0.08 nM. This work provides a powerful strategy of controlling calcination in enhancing the performance of ECL sensors, which can be used in application such as immunoassays, food safety, environmental monitoring and bioanalysis.

## Figures and Tables

**Figure 1 molecules-27-06863-f001:**
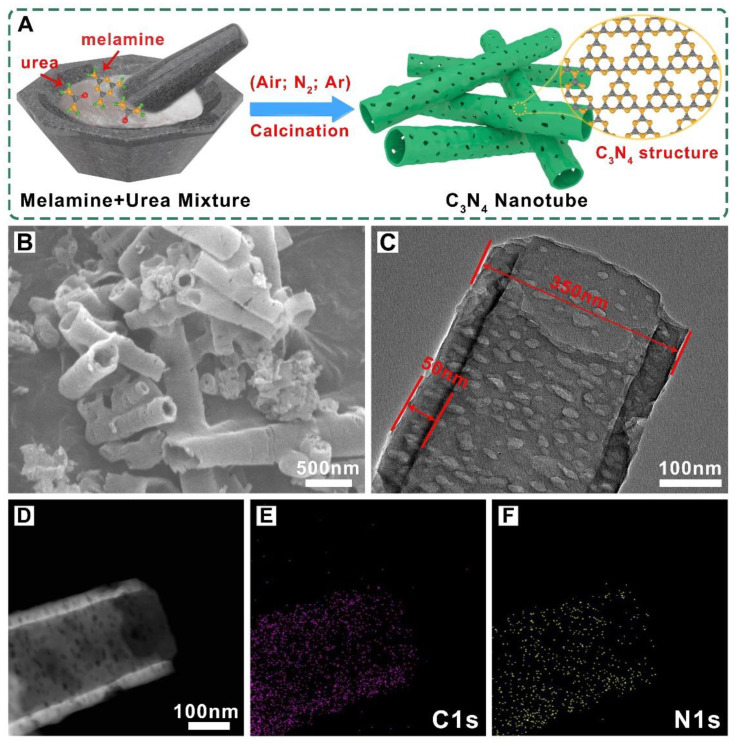
(**A**) The fabrication process of the C_3_N_4_ nanotube at different atmosphere. (**B**,**C**) The SEM and TEM images of C_3_N_4_ nanotube calcination in Air atmosphere. (**D**,**E**,**F**) High-angle annular scanning TEM images of the C_3_N_4_ NT-Air and the corresponding EDS mapping for the C and N elements.

**Figure 2 molecules-27-06863-f002:**
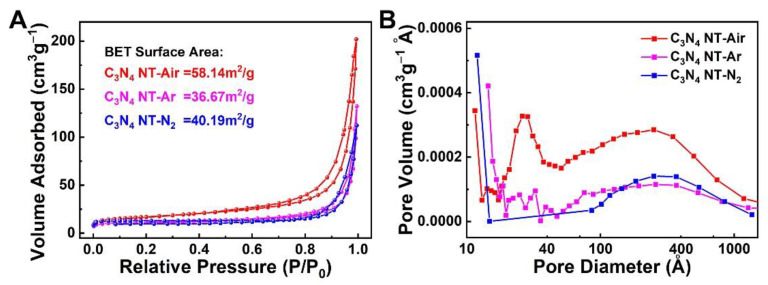
(**A**) Nitrogen adsorption-desorption isotherms of C_3_N_4_ nanotubes prepared at different atmospheres and (**B**) the corresponding BJH pore sizes.

**Figure 3 molecules-27-06863-f003:**
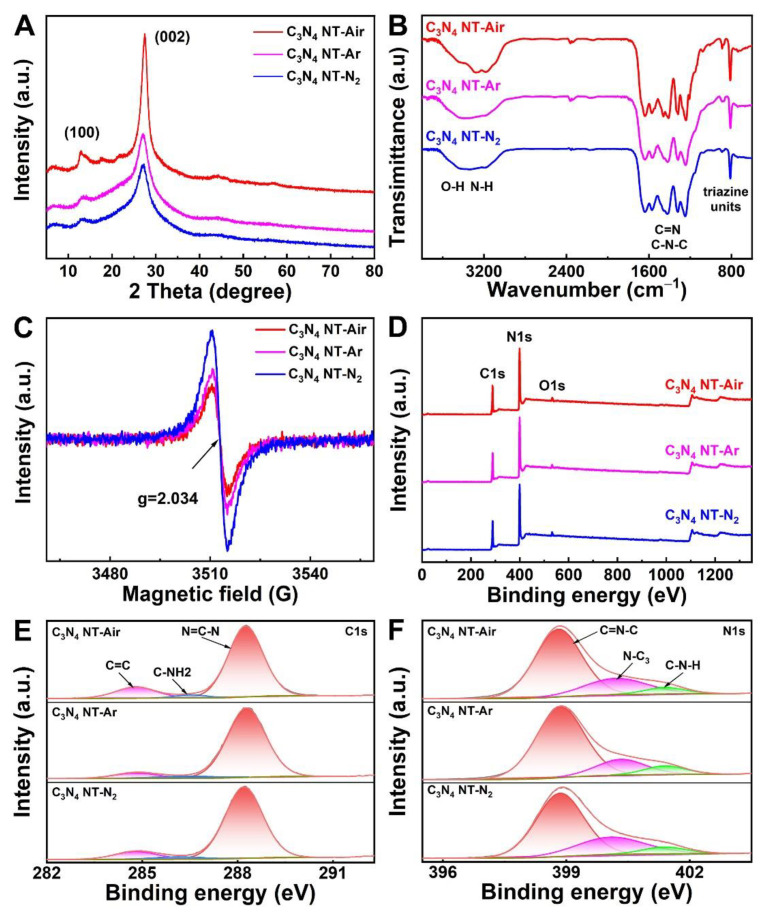
(**A**) XRD patterns, (**B**) FT-IR spectra and (**C**) EPR spectra of C_3_N_4_ NT-Air, C_3_N_4_ NT-Ar and C_3_N_4_ NT-N_2_. (**D**) The survey XPS spectra, (**E**) C1s and (**F**) N1s XPS spectra of samples.

**Figure 4 molecules-27-06863-f004:**
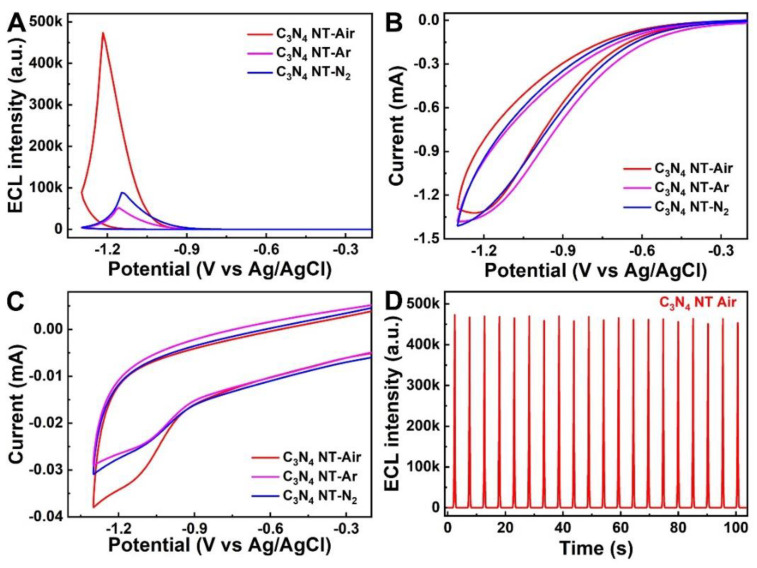
(**A**) ECL response and (**B**) CV curves of C_3_N_4_ NT-Air, C_3_N_4_ NT-Ar and C_3_N_4_ NT-N_2_ modified GCE in 0.1M PBS with 100 mM K_2_S_2_O_8_. (**C**) CV curves of C_3_N_4_ NT-Air, C_3_N_4_ NT-Ar and C_3_N_4_ NT-N_2_ modified GCE in 0.1M PBS without K_2_S_2_O_8_. (**D**) Stability of the C_3_N_4_ NT-Air modified GCE ECL system for 20 scans.

**Figure 5 molecules-27-06863-f005:**
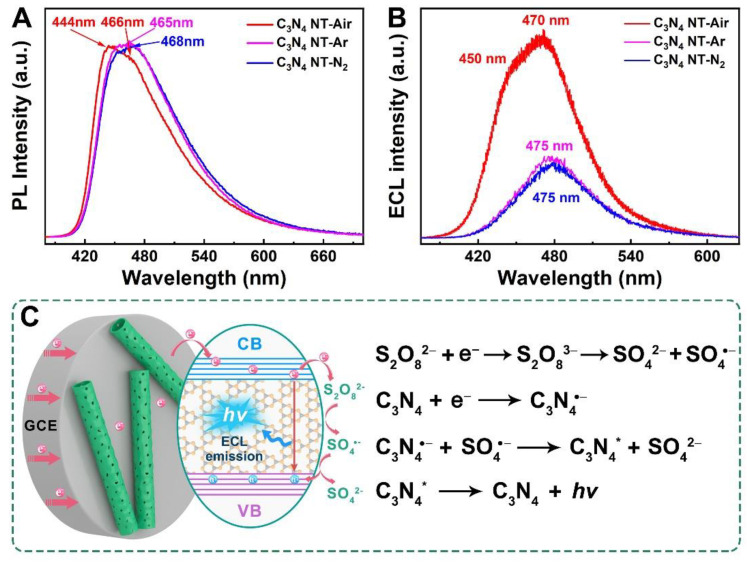
(**A**) Normalized PL spectra and (**B**) ECL emission spectra of three kinds of C_3_N_4_ nanotubes prepared at different atmospheres. (**C**) Possible ECL mechanism of the C_3_N_4_ nanotube/K_2_S_2_O_8_ system.

**Figure 6 molecules-27-06863-f006:**
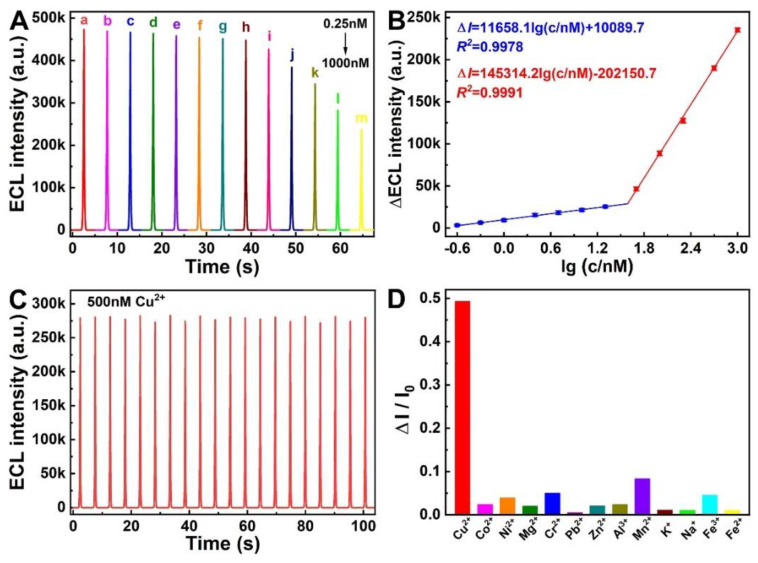
(**A**) The ECL response of C_3_N_4_ NT-Air/GCE electrodes for Cu^2+^ at different concentrations (a–m: 0.00, 0.25, 0.50, 1.00, 2.50, 5.00, 10.00, 20.00, 50.00, 100.00, 200.00, 500.00, 1000.00 nM Cu^2+^). (**B**) The calibration curve of the change of ECL intensity vs. the logarithm of Cu^2+^ concentration. (**C**) The ECL stability of the developed Cu^2+^ sensor under consecutive CV scans for 20 cycles (Cu^2+^ concentration at 500 nM). (**D**) The interferences on the Cu^2+^ detection (ECL response of the sensor towards 1 μM Cu^2+^ and 20 μM other metal ions, respectively).

## Data Availability

The data can be made available upon reasonable request.

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
