# Peer review of "A Unique, Porous C3N4 Nanotube for Electrochemiluminescence with High Emission Intensity and Long-Term Stability: The Role of Calcination Atmosphere"

_molecules, 2022, doi:10.3390/molecules27206863_

Round 1
Reviewer 1 Report
The manuscript describes the role of calcination atmosphere in tuning the properties of C3N4 nanotube and show how these materials can improve ECL performances when synthetised in optimal conditions. By providing a complete multi-analytical characterisation of the C3N4 nanotube properties upon different calcination atmospheres, this work allows a complete understanding of the correlation between structural and electrocatalytic properties. These materials are further applied in an ECL sensing strategy for Cu2+ detection. Overall, the manuscript is well-strcutured, complete and rich in contents. To imporve it, I would suggest to address/modify the following points:
1) The description of Figure 1 should be improved by discussing more the TEM images H, I and J. The magnitude of the SEM image B should be the same of image C and D. The SEM images can be moved to the S.I. since they are not underlining any meaningful difference in C3N4 nanotube structure. Overwhise the authors should add comments on the level of aggretation or clearly state that SEM images show a similarity in the nanotube structure in the um scale. Also TEM images at the same magnitude should be presented and implemented by adding arrows with average values of nanotube diameters or lenghts. These information would help the read appreciating the sturctural differences derived from the calcination atmospheres tested.
2) Please organise the text of Results and Discussion in subsection with specific titles to guide the reader through the different steps of your work. The sensing strategy is clearly explain and to improve the manuscript the results obtained should be compared with the ones of other ECL sensors previously reported.
3) Please improve the conclusion paragraph by discussing in which other sensing platform such C3N4 nanotube should be further tested.
Reviewer 2 Report
Dear authors,
the manuscript titled "A Unique-Porous C3N4 nanotube for Electrochemiluminescence with High Emission Intensity and Long-term Stability: the Role of Calcination Atmosphere" by Bolin Zhao, Xingzi Zou, Jiahui Liang, Yelin Luo, Xianxi Liang, Yuwei Zhang,* and Li Niu* is very interesting. Moreover, is well presented and done. The synthesis method is very detailled, and authors used several characterition techniques as well as interesting properties meassuremts.
The figures and graphics are of a verythey are of a high quality. On the other hand, I think that the clarity of the microphotographs, especially of figure 1, and specifically E, F and G seem to show a high sensitivity of the materials to the electron beam, since damaged areas can be observed. Have the authors attempted to do this structural study in low-dose beam conditions, to avoid damage? In addition, the contrast of figures 1 (I) and (J) I think can be improved, because in fact I can not observe its content, although the authors explain it.
The conclusions are good although they can be somewhat more detailed and complete.
